# Multifactorial assessment of leukocyte reduced platelet rich plasma injection in dogs undergoing tibial plateau leveling osteotomy: A retrospective study

Yazdan Aryazand[1¤], Nicole J. Buote[2]*, YuHung Hsieh[1], Kei Hayashi[2], Desiree Rosselli[1]

1 VCA West Los Angeles, Los Angeles, California, United States of America, 2 Department of Clinical Sciences, Small Animal Surgery Section, Cornell University College of Veterinary Medicine, Ithaca, New York, United States of America

¤ Current address: VCA Veterinary Specialists of the Valley, Woodland Hills, California, United States of America
* njb235@cornell.edu

**Data Availability Statement:** The code and data for this study is posted to https://github.com/stats-matt/PRP-study.

## Abstract

This study assessed the effects of concurrent intra-articular injection and Tibial Plateau Leveling Osteotomy (TPLO) plate surface treatment with leukoreduced platelet rich plasma (IPRP) on outcomes of dogs undergoing TPLO. A retrospective study of medical records for cases presenting from January 2018 to December 2020 was performed. Client-owned dogs with naturally occurring cranial cruciate ligament rupture that underwent TPLO surgery were divided into two groups. The IPRP group included cases that underwent intra-articular injection and plate surface treatment at the time of their TPLO. The control group (C) underwent TPLO without PRP treatment. Data analyzed included: presence of surgical site infection, implant removal rate, degree of change in OA progression score, lameness score progression and radiographic bone healing. The short- and long-term complication rate, hospitalization and antibiotic therapy were also compared between the groups. Descriptive statistics, comparison analyses (Chi square test, t-test, Fisher's exact test) and multi-level logistic regression models were used for statistical analysis. A total of 110 cases met the study inclusion criteria: 54 = IPRP, 56 = C. There were no significant differences between groups with regard to gender, age, presence of meniscal tear, weight, or body condition score. Significant findings included: improved radiographic healing of the osteotomy in the IPRP group, improved global OA scores in the IPRP group, and improved lameness score at recheck examination in the IPRP group. There was no significant difference between the IPRP and C group with regard to surgical site infection and implant removal rate. Concurrent intra-articular injection and plate surface treatment with leukocyte reduced PRP at the time of TPLO, is beneficial in slowing the progression of OA, hastening the radiographic evidence of osteotomy healing, and improved lameness score on recheck examination. Leukocyte reduced PRP was not a significant factor in reducing SSI or implant removal rate.

**Funding:** The authors received no specific funding for this work.

**Competing interests:** The authors have declared that no competing interests exist.

## Introduction

Cranial cruciate ligament (CrCL) rupture is the most common injury to the canine stifle and the leading cause of lameness in dogs [1]. Although there are variety of treatment options available to address CrCL rupture in dogs, Tibial Plateau Leveling Osteotomy (TPLO) remains the most commonly performed surgical procedure in the United States [2]. While TPLO generally has excellent predictable outcomes [3], there is evidence that osteoarthritis (OA) still progresses over time [4, 5]. Stabilization of the CrCl deficient stifle has been reported to slow, but not halt the progression of OA [5–7].

Infection is one of the most common complications following TPLO. The surgical site infection rate following TPLO ranges 3–15% [8–11]. Increased infection rate following TPLO may be due to a number of factors such as: surgeon's experience, breed, duration of anesthesia, and performing bilateral simultaneous TPLO [11]. Reported TPLO implant removal rate due to infection is 3.5–7.5% [11, 12]. The sequelae of surgical site infection or implant removal surgery include increased owner dissatisfaction, financial cost, and patient morbidity [2, 12]. A variety of strategies have been proposed to decrease TPLO post-operative infection rate, such as the use post-operative antibiotics, the use of strict shaving and prepping protocols, and adhesive drapes (IO-ban surgical adhesive 3M, Canada) [13]. The use of postoperative antibiotic therapy has been debated as a factor that could decrease TPLO infection rate [9, 10] though a recently published literature review, found little evidence to support this strategy [14].

Platelet rich plasma (PRP) has been demonstrated to have antibacterial activity including against methicillin-resistant infected skin wounds in dogs [15, 16]. PRP is created after centrifugation of autologous blood and contains a platelet concentration of at least 3–5 times that of the patient's peripheral blood [17]. Intra-articular PRP injections have also shown favorable results in reducing inflammation following CrCL rupture in dogs [18]. PRP has been shown to possess anti-inflammatory properties [19] and works as a stimulator of bone healing [20, 21]. Interestingly, intra-articular injections of PRP have been effective in slowing the progression of osteoarthritis (OA) in both humans [22, 23] and dogs [24], particularly in the canine stifle [18, 24].

The commercially available veterinary PRP products can be classified based on leukocyte count. Leukocyte increased products may elicit an inflammatory reaction following administration because leukocytes produce primarily pro-inflammatory cytokines [19, 25]. Leukocyte reduced (leukoreduced) PRP products, on the other hand, contain negligible number of leukocytes and are therefore less likely to cause a pro-inflammatory reaction [18, 19, 26]. Leukoreduced PRP injections modulate inflammation in joints by stimulating anabolism in the tissues, reducing catabolism, and enhancing viscoelastic properties [25–27]. Clinically, multiple injections of leukoreduced PRP in a CrCL deficient stifle, have been proven to improve pain and lameness scores in dogs [28].

Veterinary literature on the effects of PRP injection following CrCL rupture is currently limited. One study evaluated proximal tibial osteotomy healing when PRP was injected into the osteotomy at the time of surgery and found no significant difference in osteotomy healing time [29]. Another study investigated the effects of multiple intra-articular PRP injections on pain scores and functional outcome in an experimental CrCL deficient model in dogs. This study demonstrated improved scores for both variables with PRP injections [28]. To the author's knowledge, there is no study assessing either surgical site infection rates or OA progression in stifles receiving intraoperative leukoreduced PRP (lPRP) injection at the time of TPLO in dogs.

The objectives of our study were to assess the effects of concurrent intra-articular injection and TPLO plate surface treatment with leukocyte reduced platelet rich plasma (lPRP), at the

time of TPLO, on surgical site infection rate, implant removal rate, OA progression score, lameness score and radiographic bone healing. We hypothesized that dogs treated with lPRP at the time of TPLO would: have lower incidence of surgical site infection, lower implant removal rate, decreased OA progression score, improved lameness score and equivalent osteotomy healing compared to controls.

## Material and methods

### Study design: Case-control retrospective case series

Dogs undergoing TPLO for CrCL insufficiency from January 2018 to December 2020 were retrospectively included in our data set. Medical records were reviewed, and cases were divided into two groups based on surgeon preference and owner wishes. The lPRP group (lPRP) consisted of patients receiving intra-articular injection of lPRP (autologous conditioned plasma by Arthrex, Naples, Florida) as well as TPLO plate surface treatment. The control group (C) patients underwent TPLO without PRP injection or plate surface treatment. This study received VCA administrative approval but did not require full IACUC approval as it was retrospective in nature and previously collected data may be used in retrospective studies without IACUC approval.

### Inclusion/Exclusion criteria

Inclusion criteria was any dog undergoing TPLO regardless of meniscus status within the mentioned time frame that had not previously received any PRP products. A minimum of 6 months follow-up was required for inclusion. Any long-term complication, incision infection, implant removal, persistent lameness, and whether a contralateral TPLO was performed during the follow up period was documented. Staged bilateral TPLOs were included in both groups.

Exclusion criteria consisted of bilateral simultaneous TPLO procedures, dogs undergoing any other orthopedic procedure concurrently, dogs diagnosed with any non-CrCL related orthopedic or neurologic condition affecting gait, incomplete medical records, and dogs without examination follow up (either at our hospital or the primary veterinary office) for at least 6 months.

### Data collection

Surgical and patient data retrieved from the medical records included: sex, age, date of TPLO procedure, duration of hospitalization, body weight in kilograms, body condition score (BCS, scale ranging from 1 = emaciated, 5 = ideal, to 9 = morbidly obese) [30], affected limb (right or left), visual lameness score prior to surgery (ranging from grade 0 = no lameness noted, grade 1 = intermittent, mild weight bearing lameness, grade 2 = obvious, moderate, weight bearing lameness, grade 3 = toe touching lameness, grade 4 = Non-weight bearing lameness) [31], pre-surgical tibial plateau angle (TPA), Pre-surgical platelet count of lPRP g, post-surgical TPA, subjective intra-operative synovitis assessment (mild, moderate, severe), intra-operative OA assessment (mid, moderate, severe), TPLO plate size and type (3.5mm broad, 3.5mm standard, 3.5mm mini, 2.7mm), volume of PRP injected (if noted), administration of local injection of bupivacaine liposome injectable suspension (NOCITA, Elanco Animal Health, Greenfield, IN, USA), intra-operative complications, and post-operative oral antibiotic therapy. All preoperative and postoperative TPA's were measured by one author (YA). Available pre and recheck follow up radiographs were interpreted by a board-certified radiologist (YH) and OA scores

were assigned to each radiograph. Each radiograph was assigned two different OA scores based on previously published literature [32, 33].

Data collected from follow up visits included: appearance of the incision or any short-term complications at the 2 week recheck/suture removal examination, lameness score at the time of suture removal and at the time of recheck radiographs (on average 6–10 weeks postoperatively), any long-term complications (up to 2 years following the procedure), and whether a bilateral staged TPLO was performed during the study period.

Short term complications were considered those occurring from time of surgery to suture removal and long-term complications were considered those occurring from suture removal to the last available follow up. Complications included: signs consistent with surgical site infection (SSI) during the entire follow up period, need for implant removal, persistent lameness, or any implant complications or procedure related fractures.

## Surgical procedures

All cases were anesthetized using protocols determined by the primary clinician or board-certified anesthesiologist which included premedication with hydromorphone 0.1mg/kg IV, induction with propofol titrated to effect 6mg/kg IV, and isoflurane maintenance. Patients also received cefazolin 22mg/kg IV every 90 minutes for the duration of surgery and then every 8 hours for 24 hours post-operatively. All dogs underwent TPLO by a board-certified small animal surgeon or a surgery resident under direct supervision of a board-certified surgeon. The decision to use lPRP was based on surgeon's preference and owner financial ability. Meniscal debridement via caudal pole hemimeniscectomy was performed only if a meniscal tear was diagnosed. Meniscal release was not performed in any of the groups. LPRP was administered upon closure of the arthrotomy, and the volume recorded. All plates were Depuy Synthes locking TPLO plates (Synthes, Philadelphia, USA).

## Post-operative care, hospitalization and discharge

All cases were hospitalized for at least 24 hours following surgery. Most cases were discharged within 24 hours from the time of surgery, and if they stayed longer, it was due to owner preference. Unless contraindicated, all dogs were discharged with appropriate dose of non-steroidal anti-inflammatory medications for 10 days and gabapentin 5–10 mg/kg orally every 8–12 hours for 7–14 days. All incisions were covered with an adherent covering (Primapore, Smith +New, Watford, England, UK) while in hospital. At the time of discharge, no patient had an incision cover or bandage in place. Some cases from both groups were discharged on a 7-day course of oral antibiotics (Cefpodoxime 5–10 mg/kg PO once daily) based on surgeon preference.

## Leukoreduced PRP preparation

LPRP (ACP) was prepared following previously published guidelines [34, 35]. Briefly, once dogs were premedicated for surgery, blood was obtained using aseptic technique with either an 18-gauge butterfly catheter placed in the jugular vein or from cephalic vein during intravenous catheter (IVC) placement. Approximately 10-15mls of blood was collected in the provided double syringe mechanism (Arthrex incorporations, Naples, Florida). The blood was centrifuged at 1500 revolutions per minute (rpm) for 5 minutes (Hettich Rotofix 32, Arthrex Inc.). Centrifugation separates the red blood cells and majority of the white blood cells into a separate compartment of the syringe from the plasma. Total lPRP preparation time was approximately 20 minutes. With this system, no anticoagulant citrate is used.

### lPRP injection & treatment

The lPRP was transferred into a sterile syringe intra-operatively and injected into the stifle joint (1.5-3ml depending on the amount available) upon closing the arthrotomy. The remaining lPRP (approximately 0.5-1ml) was applied over the osteotomy, and TPLO plate after copious lavage and prior to closure of the fascial layer.

### Recheck examination & suture removal

Approximately 10–14 days post-surgery, all cases returned for a recheck examination and suture removal. Appearance of the incision, lameness score, any short-term complications (those occurring from surgery to the suture removal), and any evidence of surgical site infection, were documented. If any incisions showed evidence of purulent discharge concerning for incision infection, an aerobic culture and sensitivity was recommended.

### Recheck radiographs

A recheck exam with sedated radiographs to assess osteotomy healing was recommended 6–10 weeks post-surgery (unless there were any complications in which case, patients were evaluated earlier). Lameness score, appearance of the incision, long term complications (those occurring from suture removal to last available follow up), radiographic evidence of osteotomy healing, stability of the stifle under sedation and any other complications were documented.

### Radiographic OA score

All available pre-operative and follow up recheck radiographs were assessed by a single board-certified radiologist (YH) and two different OA scores assigned to each radiograph (as defined below). The radiologist was blinded to the groups. The OA scores were selected to provide two different OA scores for each radiograph [32, 33].

The objective OA scoring system (32) consisted of: Grade 0 = normal/no OA, effusion or osteophytes, grade 1 = early OA, stifle effusion only; no osteophytes present, grade 2 = mild OA, osteophytes on patella and femoral trochlea ridges only, grade 3 = moderate OA, small osteophytes on patella, femoral trochlea ridges, femoral condyles, fabellae, periarticular margins of the tibial plateau, and fibular head only, grade 4 = moderate to severe OA, medium to large osteophytes on patella, femoral trochlea ridges, femoral condyles, fabellae, periarticular margins of the tibial plateau, and fibular head only; mild to moderate subchondral sclerosis, and grade 5 = severe OA, osteophytes on patella, femoral trochlear ridges, femoral condyles, fabellae, periarticular margins of the tibial plateau, fibular head, and within the intercondylar notch; marked calcification and subchondral sclerosis.

The subjective OA score [33] describes the following: Global score for overall disease severity (grade 0 to 3). The global score was a summation of these categories: joint effusion (grade 0 to 2), osteophytosis (grade 0 to 3), intra-articular mineralization (grade 0 to 2) and tibial subchondral sclerosis (grade 0 to 1).

### Surgical site infection (SSI), antibiotic therapy and implant removal

The following physical examination findings were considered consistent with a surgical site infection based on previously published guidelines [36, 37]: a) Purulent drainage from the incision, b) Bacteria aseptically cultured from the incision site, c) Presence of a draining tract, d) Heat, redness, pain, or localized swelling) Incision reopened by surgeon unless there were negative culture results.

Redness, inflammation and swelling that did not lead to incision opening and a positive culture, was not considered as surgical site infection, and was categorized as inflamed incision. The primary investigator (YA) reviewed all records and determined categorization for clinical SSI or inflamed.

Any implant removal was recorded, and any culture result documented. Dogs from either group that were prescribed oral antibiotics at discharge, were excluded from this part of the study.

## Statistical analysis

**Descriptive statistics.** Data was assessed for normality with a Shapiro- Wilk test. Descriptive statistics were reported as frequency for each category (percentage), mean +/- standard deviation (SD) for normally distributed data. Medians were used where data was not normally distributed. All counted values except mean pre-op and post-op TPA were done using binomial tests (against an even split of 50%-50%). The mean pre-op and post-op TPA range was done using a two-sample t-test.

**Determining differences between PRP and non-PRP groups.** To ensure that the PRP and non-PRP groups were comparable, tests were conducted on various attributes. To determine that groups were not significantly different with regards to gender or the presence of a meniscal tear, chi-square tests were run. To determine that the groups were not significantly different with regards to age or weight, t-tests were run. To determine with there was no significant difference with regards to BCS, a Fisher's exact test was run.

**Modeling effects of PRP.** To investigate the effects of PRP and other variables on the outcomes of interest, regression models were created. Because some dogs were represented multiple times within these data, multi-level regression models were created using patient as a random effect. For the dichotomous outcome variables of worsening of OA (for all aspects of OA assessment such as Effusion score, Global score, osteophyte score), multi-level logistic regression was used. Predictors of PRP group, gender, age, weight, BCS, and presence of a meniscal tear were included as potential variables in the model to determine which were significant. To compare the lengths of hospital stays, a linear mixed model was created which included the same list of potential predictors as the logistic models above. To compare effusion change scores, a multinomial regression model was created where the outcome variable was categorized as being better, worse, or the same. To avoid assumptions about the relationships between these three levels and the predictors, effusion change was not considered to be an ordered variable.

The code and data are posted to https://github.com/stats-matt/PRP-study.

## Results

### Signalment data

A total of 110 cases met the study inclusion criteria with 54 TPLOs in the lPRP group and 56 in the control (C) group. The median age for all cases was 6 years old (1–13 years). The median age for the C group was 6 years (1–13 years) and the median age for the lPRP group was also 6 years old (2–11 years). The population of the dogs that received lPRP and those that did not receive lPRP were not significantly different. There were no significant differences in these populations with regard to gender (p = 0.17), age (p = 0.59), presence of meniscal tear (p = 0.39), weight (p = 0.06), or BCS (p = 0.67). Forty-seven dogs were castrated male, 60 were female spayed, two male intact and one female intact. Dogs were of a variety of breeds with Pitbull being the most common breed (35 cases). Other represented breeds included: German shepherds (12), Labrador Retriever (12), English bulldog (8), mix breed (7), golden retriever

(7), Boxer (4), Alaskan Malamute (3), American Staffordshire terrier (3), husky (2), Newfoundland (2), Rottweiler (2), sheepdog (2), Akita (2), Doberman Pincher (2), Mastiff (1), cocker spaniel (1), standard poodle (1), pointer (1), and whippet (1). Mean body weight for all dogs was 29.2 kilograms (kg). Mean body weight was 28.3 and 29.7 kg for C and lPRP group, respectively. Mean body condition score (BCS) was 5 (out of 9) for all cases. Mean BCS for each group was 5 out of 9.

## Surgical data (Table 1)

The results of TPLO leg, pre-op platelet count, pre-op TPA, post-op TPA, meniscal status, Liposomal Bupivacaine injection, intra-op synovitis, intra-op OA assessment, TPLO plate type and intra-op complications are summarized in Table 1.

Briefly, 45 cases had right TPLO, 33 had left TPLO, and 16 cases underwent bilateral staged TPLO between both groups. In the C group, 22 cases had right TPLO, 14 had left TPLO, and 10 cases had bilateral staged TPLO. In the lPRP group, 23 cases had right TPLO, 19 cases had left TPLO, and 6 dogs had bilateral staged TPLO. The mean preoperative platelet count for the lPRP group was within normal reference range at 236. 9k/μL (SD = 59.4). Mean pre-operative TPA for all cases was 28.3 degrees. Mean pre-operative TPA were 27.8 and 28.7 degrees in C and lPRP group, respectively. There was no significant difference in TPA between groups (Two sample t-test, p = 0.077). Mean post-op TPA was 7.68 (SD = 2.75) and 6.18 (SD = 2.27) for the C and lPRP groups respectively (p = 0.003). Sixty cases received local injection of bupivacaine liposome injectable suspension at 0.4 ml/kg and 50 cases did not. In the C group 40 cases received liposomal bupivacaine injection upon closure of the fascia while 16 cases did not. In the lPRP group, 20 cases received liposomal bupivacaine injection and 34 cases did not.

**Table 1. Summary of surgical data for all cases.**

| Variable | Overall | Control | PRP | P value |
|---|---|---|---|---|
| Right TPLO | 45 | 22 | 23 | 1.00 |
| Left TPLO | 33 | 14 | 19 | 0.487 |
| Bilateral staged TPLO | 16 | 10 | 6 | 0.454 |
| Mean Pre-op TPA (range) | 28.3 (23–45) | 27.8 (23–32) | 28.7 (23–45) | 0.077 |
| Mean post-op TPA (range) | 6.9 (1–13) | 7.7 (1–13) | 6.2 (1–13) | 0.003 |
| Meniscal tear | 43 | 19 | 24 | 0.542 |
| Liposomal bupivacaine injection | 60 | 40 | 20 | 0.013 |
| Intra-op synovitis (reported) | 69 | 47 | 22 | 0.004 |
| Mild | 39 | 32 | 7 | <0.001 |
| Moderate | 26 | 11 | 15 | 0.557 |
| Severe | 4 | 4 | 0 | 0.125 |
| Intra-op OA (reported) | 53 | 30 | 23 | 0.41 |
| Mild | 15 | 8 | 7 | 1.00 |
| Moderate | 34 | 19 | 15 | 0.608 |
| Severe | 4 | 3 | 1 | 0.625 |
| TPLO plate type | 110 | 56 | 54 | 0.924 |
| 3.5mm mini | 21 | 16 | 5 | 0.027 |
| 3.5mm standard | 69 | 34 | 35 | 1.00 |
| 3.5mm broad | 19 | 5 | 14 | 0.064 |
| 2.7mm | 1 | 1 | 0 | 1.00 |
| Intra-op complication | 10 | 3 | 7 | 0.344 |

Among the 110 cases, 66 had intact meniscus, while 43 were diagnosed with a medial meniscal tear. One surgery report did not comment on the status of the meniscus. In the C group, 36 menisci were diagnosed to be intact while 19 medial meniscal tears were diagnosed (one did not comment on the status of the meniscus). In the lPPR group, 30 menisci were intact and 24 were diagnosed with medial meniscus tear. Subjective intra-operative synovitis and OA were also documented in each group. Overall, intra-operative synovitis was documented in 69 surgery reports, with 39 cases being mild, 26 moderate and 4 severe synovitis. In the C group, 32 cases had mild intra-op synovitis, 11 had moderate synovitis, and 4 cases had severe intra-op synovitis. In the PRP group, 7 had mild synovitis, 15 had moderate synovitis and no case of severe intra-operative synovitis was noted. Statistically there was a significant difference in intraoperative assessment of synovitis, with more cases with synovitis reported in the control group compared to the PRP group. Intra-operative OA was documented in 53 cases overall with 15 being mild, 34 being moderate and 4 being severe. In the C group, 8 cases had mild OA, 19 had moderate and 3 had severe. In the PRP group, 7 had mild, 15 had moderate and 1 case had severe OA.

## Surgical site infection, incision appearance, lameness scores, complications at suture removal, and plate removal rate

Within the 110 cases recruited in the study, 31 cases were prescribed antibiotics at discharge. These cases were excluded from the infection-related data analysis. Out of 79 cases that were not discharged with oral antibiotics, 42 were in the C group while 37 were in the lPRP group. The results are summarized in Table 2.

**Table 2. Summary of post-operative lameness score assessment, osteotomy healing, complications and SSI.**

| Variable | Overall | Control | PRP | P value |
|---|---|---|---|---|
| Lameness score (0–4) at suture removal | 108 | 55 | 53 | 0.923 |
| 0 | 1 | 0 | 1 | 1.00 |
| 1 | 51 | 18 | 33 | 0.049 |
| 2 | 48 | 33 | 15 | 0.013 |
| 3 | 7 | 4 | 3 | 1.00 |
| 4 | 1 | 0 | 1 | 1.00 |
| Lameness score (0–4) at recheck radiographs | 93 | 46 | 47 | 1.00 |
| 0 | 63 | 26 | 37 | 0.207 |
| 1 | 25 | 17 | 8 | 0.108 |
| 2 | 4 | 3 | 1 | 0.625 |
| 3 | 1 | 0 | 1 | 1.00 |
| 4 | 0 | 0 | 0 | NA |
| Incision appearance at suture removal | | | | |
| Infected | 4 | 3 | 1 | 0.625 |
| Inflamed | 14 | 9 | 5 | 0.424 |
| Swollen | 18 | 12 | 6 | 0.238 |
| Did not comment | 2 | 1 | 1 | 1.00 |
| Normal | 72 | 31 | 41 | 0.289 |
| Complication at suture removal | 16 | 8 | 8 | 1.00 |
| Complication at recheck radiographs | 26 | 13 | 13 | 1.00 |
| SSI | 16 | 10 | 6 | 0.455 |
| TPLO explant surgery | 8 | 2 | 6 | 0.289 |

Among these cases at suture removal appointment, 4 incisions appeared infected, 14 incisions looked inflamed, 18 incisions were swollen, two had no comment on the incision appearance and 72 incisions had healed normally. The P-values are reported in Table 2. Overall, 16 SSIs were diagnosed from the time of suture removal until the last available follow up (at least 6 months post-surgery). In the C group, 10 cases developed surgical site infection and in the lPRP group 6 cases developed surgical site infection. Implant removal surgery was recommended for all cases. Two cases in the C group underwent TPLO implant removal and 6 cases in the lPRP group had the TPLO implants removed. The remaining cases did not undergo implant removal due to financial reasons or owners deciding against putting their pet through another procedure.

## Osteoarthritis and osteotomy healing

The results of pre-operative OA scores, radiographic healing of the osteotomy and comparison between the groups are summarized in Table 3.

Changes in OA score were classified as either progressive or not progressive. If the global OA score increased after the surgery, the dog was considered to have progressive OA. If the score stayed the same, it was considered not progressive. If the post-surgery score was decreased compared to the pre-surgery score, the dog was considered not progressive. The number of dogs with progressive OA in the C group (4) was twice that of the lPRP group (2). The degree of changes in the OA scores between pre and post radiographs was compared between the groups and p-values reported in Table 3. There was no statistical difference in OA score changes between the groups (p = 0.072). For the degree of global OA score change comparison between the C and lPRP group, the lPRP group had significantly less progression of their global OA scores compared to the C group when pre- and post-op global OA scores were considered (p = 0.033). A line graph showing the average change in outcomes of global OA for dogs with and without lPRP is depicted in Fig 1.

**Modeling change in osteophytosis scores between pre-op and post-op and comparison between groups.** A total of 10 dogs had a progressive osteophytosis score (OS score) on post-op radiographs compared to their pre-op: 6 among the C and 4 among the lPRP group. While there was no significant difference in pre and post-op OS score progression between the two groups (p = 0.277), there was a significant difference between pre and post-op OS score progression between dogs that had an intact medial meniscus, compared to dogs that had a medial meniscal tear (p = 0.024).

**Table 3. Summary of post-operative OA scores and radiographic evidence of osteotomy healing.**

| Variable | Overall | Control | PRP | P value |
|---|---|---|---|---|
| Mean pre-op OA score (range 0 to 5) | 2.4 | 2.25 | 2.6 | 0.049 |
| Mean post-op OA score (range 0 to 5) | 2.7 | 2.6 | 2.8 | 0.381 |
| Mean pre-op global OA score (range 0 to 3) | 1.25 | 1.25 | 1.3 | 0.796 |
| Mean post-op global OA score (range 0 to 3) | 1.3 | 1.4 | 1.25 | 0.176 |
| Mean pre-op effusion OA score (range 0 to 2) | 1.7 | 1.6 | 1.8 | 0.103 |
| Mean post-op effusion OA score (range 0 to 2) | 1.6 | 1.5 | 1.7 | 0.163 |
| Mean pre-op osteophytosis OA score (range 0 to 3) | 1.2 | 1.1 | 1.28 | 0.107 |
| Mean post-op osteophytosis OA score (range 0 to 3) | 1.3 | 1.3 | 1.3 | 0.976 |
| Mean pre-op mineralization OA score (range 0 to 2) | 0.3 | 0.4 | 0.25 | 0.129 |
| Mean post-op mineralization OA score (range 0 to 2) | 0.45 | 0.6 | 0.3 | 0.026 |
| Osteotomy healing at recheck radiographs | 62 | 22 | 40 | 0.03 |

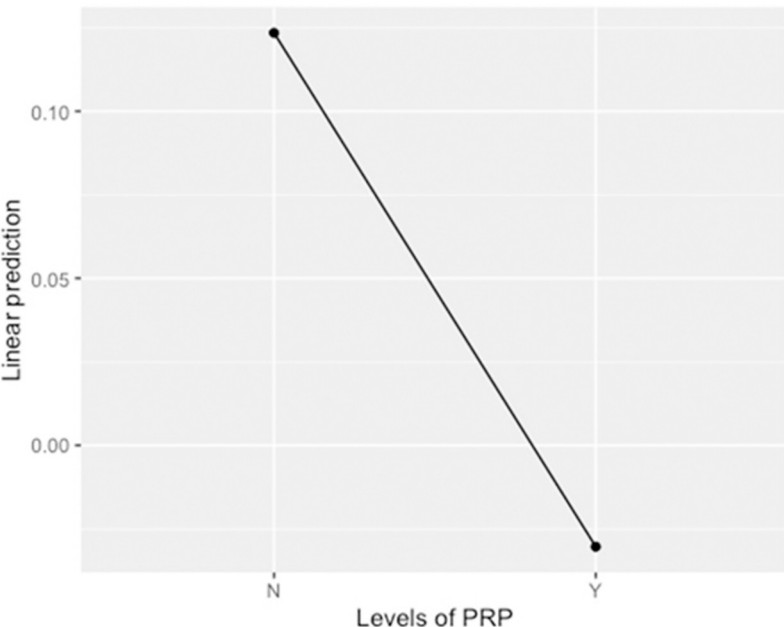

**Fig 1. Line graph showing the predicted average change in outcomes of global OA for each of the PRP and non-PRP groups.** N = not administered PRP, Y = administered PRP.

**Modeling change in pre-op and post-op mineralization scores and comparison between groups.** The degree of mineralization score change (comparing pre-op to post-op mineralization scores) was not significantly different between the two groups (p = 0.659) though the probability of mineralization progressing in the lPRP group was lower compared to the C group (p = 0.18).

**Degree of change for effusion scores and comparison between groups.** For dogs in the C group, 66% of the effusion scores stayed the same without any progression while 23% had less effusion score (lower numbers) and 11% had increased effusion scores (higher numbers). For dogs in the lPRP group, the vast majority (80%) stayed the same and there was no progression of effusion score, while 10% had decreased effusion score and the other 10% had increased effusion scores. This difference between groups was not statistically significant (p = 0.225).

**Radiographic evidence of osteotomy healing.** Radiographs were assessed 6–10 weeks post-surgery (on average 8 weeks). There was no significant difference in the timing of assessment between treatment groups. Overall, 62 cases had evidence of osteotomy healing at the time of recheck radiographs (assessed by board certified radiologist). In the C group, 22 cases had evidence of osteotomy healing while 40 cases in the PRP group had evidence of complete osteotomy healing. Radiographic evidence of osteotomy healing was compared between the two groups: Both PRP and age were significant predictors of osteotomy healing (p = 0.003 and p = 0.049, respectively.) For the average aged dog (5.84 years), the probability of radiographic healing for dogs in the C group was 51.1% compared to dogs in the lPRP group which was 87.6%. This indicates that as age increases, the probability of healing decreases, but for all ages, the probability of healing is higher for dogs in the lPRP group compared to dogs in the C group. The relationship between healing, age, and PRP can be seen in Fig 2.

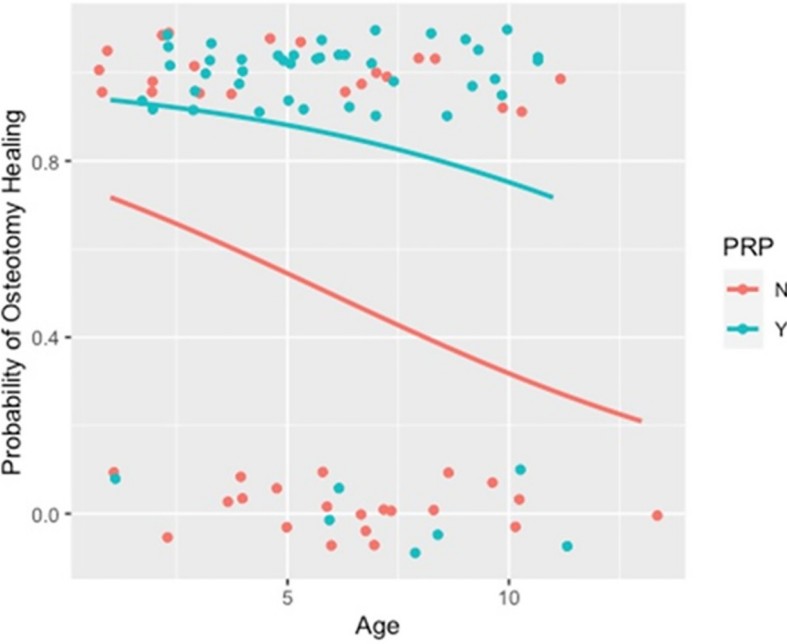

**Fig 2. Scatterplot with curves represent probability of osteotomy healing by age for each of the PRP and non-PRP groups.** N = not administered PRP, Y = administered PRP.

## SSI

There is no indication for a relationship between risk of infection in the lPRP group compared to C group. (P≤0.5, one tailed proportion test). Dogs from either group that were discharged on oral antibiotics were excluded from this part of data analysis. If the cases that were discharged on antibiotics were to be included in the study, then there would be significantly less infections in the lPRP group compared to C group (p = 0.012, one tailed proportion test).

## Lameness score at recheck exam and comparison between groups

When comparing the change in lameness score at from suture removal to score at recheck radiographs examination (6–10 weeks post-operative), dogs in the lPRP group had significantly lower lameness scores compared to dogs in the C group (p = 0.003). Having received PRP treatment, resulted in a decrease of lameness score by 0.412 on average when comparing pre-op and post-op lameness scores in dogs of the lPRP group.

## Discussion

The purpose of this study was to evaluate the effects of concurrent PRP injection at the time of TPLO and compare SSI rate, implant removal rate, degree of change in OA progression score, lameness score progression and radiographic bone healing. We found that PRP treatment significantly improved radiographic healing of the osteotomy, improved global OA scores and improved lameness score at recheck examination. We found no significant difference between the lPRP and C group with regard to SSI and implant removal rate.

Our groups in this study were homogenous with regards to age, weight, breed, pre-op TPA, the affected limb and meniscus status at the time of surgery. Postoperative TPA was found to be significantly different between groups which could, in theory, have affected postoperative

outcomes such as lameness score and possibly osteoarthritis scores as the lPRP group TPA was closer to the published preferred angle of 5°. While statistically significant, the mean difference was only 1.5°, which is within the intraobserver error published previously [38] and the authors feel this difference is likely to be clinically insignificant. Even though there was a significant difference in intraoperative assessment of synovitis, with more mild synovitis in the control group compared to the PRP group this grading score was a subjective assessment, therefor this finding is unlikely to be of clinical importance. Future studies could evaluate the postoperative outcomes of lPRP compared to more objective intraoperative synovial biopsies.

We rejected our hypothesis that PRP would have no effect on osteotomy healing. This contrasted with the findings of Franklin et al. [29] where PRP administration did not improve proximal tibial osteotomy healing. This finding could be due to multiple factors. Firstly, our larger case numbers could have played a role (54 dogs per group in our study compared to 27 per group in Franklin's study). Secondly, criteria for healing were different between the two studies. We evaluated osteotomy healing solely on plain radiographs, whereas plain radiography, ultrasound and MRI were evaluated in the Franklin study. We evaluated radiographic osteotomy healing on average 8.34 weeks post-surgically, while radiographic osteotomy healing was assessed at 28, 49, and 70 days in the Franklin study. This could affect the documented osteotomy healing, as radiographic healing is expected to progress as more time elapses. The results of our study reveal that radiographic healing at approximately 8 weeks post-surgery was significantly better in the PRP compared to C group. Both studies found age to be a significant factor affecting osteotomy healing time concluding as age increases, a longer time to radiographic osteotomy healing should be expected.

Other studies have evaluated the effects of PRP on bone healing in canine patients. Rabillard et al. [39] looked at the effects of autologous platelet rich plasma gel and calcium phosphate biomaterials on bone healing in an ulnar ostectomy model in dogs and found no improved bone healing with PRP. Souza et. al. on the other hand, found PRP to be effective in promoting bone healing in a canine radial ostectomy gap model [40]. Gianakos et. al. performed the most robust systemic review to date on the effects of PRP in animal long bone model [21] and found that 89% of studies reported significant improvement in earlier bone healing on histologic assessment. One hundred percent of patients showed significant increase in bone formation on radiographs with the PRP and 100% showed a higher torsional stiffness for the PRP-treated defects. They concluded that PRP is beneficial as a biologic adjunct in animal long-bone models. The results of the latter two studies are consistent with the results of our study.

We employed two different OA scoring systems for this study to be as objective as possible, and to compare results using two validated systems [32, 33]. The results illustrate a lower rate of OA global score progression in cases that received lPRP compared to the C group. This is consistent with multiple other studies that showed improvement in OA with PRP administration [19, 22, 23, 27, 34, 41, 42]. These results should be interpreted with caution, due to the lack of long-term radiographic follow up however 6 months follow up is consistent with some studies [22, 42]. The authors speculate whether the global OA scores would have remained significantly in favor of PRP if long term radiographs were assessed, for example,12 or 24 months post-operatively, since OA progression is expected to slow down following stabilization of CCL deficiency, regardless of the use of PRP [4–6].

The results of our study illustrated an improved lameness score on 6–10 week recheck examination for dogs following TPLO in the lPRP group compared to dogs in the C group. The results should be interpreted with caution as clinical lameness score is subjective and varies from surgeon to surgeon. The surgeons in this study routinely used this grading system and studies have shown that intraobserver variations are within acceptable limits. One theory for

improved lameness score in lPRP group could be faster healing of the osteotomy leading to earlier load bearing and improved lameness score. The lPRP is supposed to decrease inflammation through multiple pathways and this could be the primary mechanism behind the improved lameness scores as well. Future studies using an objective means of lameness assessment such as a force plate analysis are warranted.

The incidence of SSI was not significantly different between groups, which rejects our hypothesis. There are published studies highlighting the antibacterial effect of PRP in wounds [16]. Although other studies have shown antibacterial effects of PRP on methicillin resistant wounds in dogs, our study failed to show any significant difference in implant infection rate between the lPRP and the control group after removing the cases that were discharged on oral antibiotics. Although the number of dogs in each group was small after removing the patients that were treated with post operative antibiotics: ten dogs in the C group developed SSI (4 were excluded because they were discharged with oral antibiotics), and 6 dogs in the lPPR group developed SSI (0 were discharged with oral antibiotics). The authors speculate that the insignificant results could be due to low number of cases in each group (Type II error) or differences in lPRP centrifugation methodology. Interestingly, when dogs that received oral antibiotics were not excluded from this portion of the study, the results would have shown a statistical significance for reduced rate of SSI in the lPRP group. There is conflicting evidence in the literature on the role of post-operative oral antibiotic therapy in reducing the rate of SSI following TPLO. The most recent systematic literature review study failed to find an association between post-operative antibiotic therapy and decreased risk of SSI following TPLO in dogs [13].

Implant removal rate was also not significantly different between the 2 groups which rejects our hypothesis. This is most likely due to the low number of cases that required implant removal in both groups and the fact that the recommendation to remove the TPLO plate was not pursued by all owners for financial reasons. It is also possible that some cases were lost to follow up, since implant infection can occur even years after the initial procedure.

This study has several limitations inherent to a retrospective study. Cases were excluded due to incomplete medical records, and all follow-up examinations were based on the available medical records at our hospital or records that we received from the referring veterinarians. Financial limitations of clients were the primary reason to not pursue implant removal for the cases that developed SSI, which affected our implant removal case numbers. Follow-up time was also not strictly controlled, as in most retrospectives. Some of the earlier cases had up to 48 months follow up, while the more recent cases had only 6 months of follow up.

In conclusion, concurrent intra-articular injection and plate surface treatment with leukocyte reduced PRP at the time of TPLO, is beneficial in slowing the progression of OA (compared to the control group), hastening the radiographic evidence of osteotomy healing, and improved lameness score on recheck examination. LPRP was not a significant factor in reducing SSI or implant removal rate.

## Acknowledgments

The authors would like to acknowledge Matt Thomas, PhD, for his work on the statistical analysis for this report.

## Author Contributions

**Conceptualization:** Nicole J. Buote.

**Data curation:** Yazdan Aryazand, YuHung Hsieh.

**Formal analysis:** Yazdan Aryazand, Nicole J. Buote, YuHung Hsieh, Kei Hayashi, Desiree Rosselli.

**Funding acquisition:** Kei Hayashi.

**Investigation:** Yazdan Aryazand.

**Methodology:** Nicole J. Buote, Kei Hayashi.

**Project administration:** Yazdan Aryazand.

**Resources:** Nicole J. Buote.

**Supervision:** Desiree Rosselli.

**Validation:** YuHung Hsieh.

**Writing – original draft:** Yazdan Aryazand, YuHung Hsieh, Kei Hayashi, Desiree Rosselli.

**Writing – review & editing:** Yazdan Aryazand, Nicole J. Buote.

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
