## [Decision Letter · Decision Letter 0]

5 Apr 2023

PONE-D-22-35213Multifactorial assessment of leukocyte reduced platelet rich plasma injection in dogs undergoing tibial plateau leveling osteotomy: a retrospective studyPLOS ONE

Dear Dr. Aryazand

Thank you for submitting your manuscript to PLOS ONE. After careful consideration, we feel that it has merit but does not fully meet PLOS ONE’s publication criteria as it currently stands. Therefore, we invite you to submit a revised version of the manuscript that addresses the points raised during the review process.

We look forward to receiving your revised manuscript.

Kind regards,

Mohamed El-Sayed Abdel-Wanis, Ph.D.

Academic Editor

PLOS ONE

Journal Requirements:

Reviewers' comments:

Reviewer's Responses to Questions

**Comments to the Author**

1. Is the manuscript technically sound, and do the data support the conclusions?

Reviewer #1: Yes

Reviewer #2: Yes

2. Has the statistical analysis been performed appropriately and rigorously? 

Reviewer #1: Yes

Reviewer #2: Yes

3. Have the authors made all data underlying the findings in their manuscript fully available?

Reviewer #1: Yes

Reviewer #2: Yes

4. Is the manuscript presented in an intelligible fashion and written in standard English?

Reviewer #1: Yes

Reviewer #2: Yes

5. Review Comments to the Author

Reviewer #1: The paper is very interesting and well written.

As a work on PRP, the authors may consider including data on the number of platelets in the blood of patients before surgery, which, with the centrifugation method used, would allow for the standardization of the PRP used.

Reviewer #2: This manuscript examined the effects of leukocyte reduced platelet rich plasma (lPRP) on surgical site infection (SSI), implant removal, bone healing, clinical signs, progression of OA in the dogs undergoing tibial plateau leveling osteotomy (TPLO). As the results, lPRP treatment significantly improved bone healing, global OA scores, and lameness score. However, there were no significant differences in the results of SSI and implant removal rate. The content of this paper is worthy of publication in Plos One, because of its contribution to the clinical field of veterinary medicine. However, some revisions of manuscript are needed before it can be accepted for publication.

Introduction (line 79): “1PRP” is a first appearance. Please list the term that is not abbreviations (Ex. Leukoreduced platelet rich plasma?).

M&M (line 87-93, 136-137): How did you divide the subject dogs into two groups? What were the criteria of surgeon's preference?

M&M (line 101-103): Orthopedic (excluding CrCL rupture) and neurological disorders that may affect gait and mobility should be also added to the exclusion criteria.

M&M (line 139-142): This content is described below (lines 164-167), so delete the relevant text.

M&M (line 150): pet→dog

M&M (line 151): Please describe which type of antibiotic was administered orally.

Results (line 262-270): Did you compare post-op TPA between groups? These values might affecte the outcome of the surgery.

Results (line 277-281): Add the sentence about significant difference in the presence of synovitis between groups.

Table 2: Correct any misalignments in the Table 2 (Incision appearance at suture removal).

Discussion (line 415-425): In this paragraph, you need to add a discussion of the role of PRPs in infection protection and the results of this study.

Discussion (line 438-442): leukocyte reduced PRP→lPRP?

Discussion: In this study, lameness was significantly improved in the lPRP group. You should discuss whether the accelerated bone healing in the lPRP group was the result of improved loading on the tibial bone or the effect of PRP in the Discussion.

Reference: There are two No. 29 references. Please check again, including the citation in the text.

6. PLOS authors have the option to publish the peer review history of their article (what does this mean?). If published, this will include your full peer review and any attached files.

Reviewer #1: No

Reviewer #2: **Yes: **Kazuya Edamura

---

## [Author Response · Author response to Decision Letter 0]

21 May 2023

Response to Reviewers

Review Comments to the Author

Reviewer #1: The paper is very interesting and well written.

Author response: Thank you. We appreciate your support and the time you have taken to review our manuscript.

As a work on PRP, the authors may consider including data on the number of platelets in the blood of patients before surgery, which, with the centrifugation method used, would allow for the standardization of the PRP used.

Author response: Thank you for bringing up this interesting point. We have added this information to the methods and results section for the lPRP group. We did not feel statistical analysis was necessary for the platelet count as there is no gold standard number to compare to and the data was normally distributed. Please let us know if you feel we need to add this in more detail. 

Reviewer #2: This manuscript examined the effects of leukocyte reduced platelet rich plasma (lPRP) on surgical site infection (SSI), implant removal, bone healing, clinical signs, progression of OA in the dogs undergoing tibial plateau leveling osteotomy (TPLO). As the results, lPRP treatment significantly improved bone healing, global OA scores, and lameness score. However, there were no significant differences in the results of SSI and implant removal rate. The content of this paper is worthy of publication in Plos One, because of its contribution to the clinical field of veterinary medicine. However, some revisions of manuscript are needed before it can be accepted for publication.

Author response: Thank you for your time and your suggestions. We have tried to address each of your concerns below. 

Introduction (line 79): “1PRP” is a first appearance. Please list the term that is not abbreviations (Ex. Leukoreduced platelet rich plasma?).

Author response: The first appearance of the term lPRP is actually in line 15 where the full term (Leukoreduced Platelet rich Plasma) has been listed. We have added it to line 79 as well just to be safe. 

M&M (line 87-93, 136-137): How did you divide the subject dogs into two groups? What were the criteria of surgeon's preference?

Author’s Response: The dogs were divided into groups based on surgeon’s preference. One surgeon (NJB) and one surgery resident (YA) preferred patients receive lPRP injection while other surgeons and residents did not. Not all cases of NJB or YA received lPRP though as sometimes the owners declined the cost of lPRP injection, thus those cases had routine TPLO without PRP injection. We have added this information into the methods section in both of the places you referenced.

M&M (line 101-103): Orthopedic (excluding CrCL rupture) and neurological disorders that may affect gait and mobility should be also added to the exclusion criteria.

Author response: Thank you for this suggestion. We have added this to the exclusion criteria. 

M&M (line 139-142): This content is described below (lines 164-167), so delete the relevant text.

Author Response: Thank you for your comment. We have deleted the redundant content (lines 139-142) per your suggestion. 

M&M (line 150): pet→dog

Author response: Changed pet to patient.

M&M (line 151): Please describe which type of antibiotic was administered orally.

Author response: Thank you for your comment. We have added the antibiotic name and dose. Cefpodoxime (simplicef) 5-10 mg/kg PO once daily 

Results (line 262-270): Did you compare post-op TPA between groups? These values might affect the outcome of the surgery.

Author response: Thank you for this suggestion. We did not originally compare postoperative TPAs but have done so as we agree this might affect some of the postoperative outcomes (possibly lameness score and osteoarthritis score) we looked at. When analyzed there was a statistically significant difference between lprp and control groups however this mean difference was only 1.5º which is within the published intraoberver difference for TPA (added reference 38). We have added this information and stated we do not feel such a small TPA difference would have an effect clinically.

Results (line 277-281): Add the sentence about significant difference in the presence of synovitis between groups.

Author Response: Thank you for this comment. We have added a line in the results as well as the discussion. We feel the subjectivity of this score limits its clinical value but feel like future works could evaluate outcomes based on more rigorous operative diagnostics such as biopsies. 

Table 2: Correct any misalignments in the Table 2 (Incision appearance at suture removal).

Author response: Thank you for catching this. We have corrected the misalignment. 

Discussion (line 415-425): In this paragraph, you need to add a discussion of the role of PRPs in infection protection and the results of this study.

Author response: We have added information in this section. Please let us know if you feel this needs continued supplementation. 

Discussion (line 438-442): leukocyte reduced PRP→lPRP?

Author response: Changed 

Discussion: In this study, lameness was significantly improved in the lPRP group. You should discuss whether the accelerated bone healing in the lPRP group was the result of improved loading on the tibial bone or the effect of PRP in the Discussion.

Author response: Thank you for this comment. We have added a few sentences to the discussion regarding the differences in lameness scores. 

Reference: There are two No. 29 references. Please check again, including the citation in the text.

Author response: Changed and adjusted all the references in the text accordingly.

---

## [Editor Report · Decision Letter 1]

15 Jun 2023

Multifactorial assessment of leukocyte reduced platelet rich plasma injection in dogs undergoing tibial plateau leveling osteotomy: a retrospective study

PONE-D-22-35213R1

Dear Dr. Nicole Buote

We’re pleased to inform you that your manuscript has been judged scientifically suitable for publication and will be formally accepted for publication once it meets all outstanding technical requirements.

Kind regards,

Mohamed El-Sayed Abdel-Wanis, Ph.D.

Academic Editor

PLOS ONE

---

## [Editor Report · Acceptance letter]

22 Jun 2023

PONE-D-22-35213R1 

Multifactorial assessment of leukocyte reduced platelet rich plasma injection in dogs undergoing tibial plateau leveling osteotomy: a retrospective study 

Dear Dr. Buote:

I'm pleased to inform you that your manuscript has been deemed suitable for publication in PLOS ONE. Congratulations! Your manuscript is now with our production department. 

Kind regards, 

on behalf of

Prof. Dr Mohamed El-Sayed Abdel-Wanis 

Academic Editor

PLOS ONE